# An Exploratory Study of Psychological Distress, Professional Quality of Life, Effort-Reward Imbalance, and Turnover Intention of Hospital Nurses during the COVID-19 Pandemic

**DOI:** 10.3390/healthcare11192695

**Published:** 2023-10-09

**Authors:** Suk-Jung Han, Soon-Youl Lee, Sie-Eun Kim

**Affiliations:** 1College of Nursing, Sahmyook University, Seoul 01795, Republic of Korea; 2Department of Nursing, Graduate School of Sahmyook University, Seoul 01795, Republic of Korea; v-mva@hanmail.net (S.-Y.L.); lamp19@naver.com (S.-E.K.)

**Keywords:** COVID-19, psychological distress, compassion fatigue, satisfaction, burnout, reward, personnel turnover

## Abstract

This exploratory study aimed to identify factors that may influence nurses’ turnover intentions during the COVID-19 pandemic. The data were collected during January 2023 from 250 nurses and analyzed using descriptive statistics, *t*-test, ANOVA, Scheffe, Pearson’s correlation, and multiple regression analysis. Among the sociodemographic and clinical characteristics, nursing care and working with personal protective equipment significantly impacted the turnover intention. Among the independent variables, compassion satisfaction, burnout, effort–reward ratio, and psychological distress were significant, with an explanatory power of 43.3%. Among the subjects, 86.4% (216 people) showed a moderate or high burnout level because of the COVID-19 pandemic, and burnout seemed to have a significant impact on turnover intention. Therefore, to lower the turnover intention of nurses, burnout should be prevented, and managers should create an environment where nurses can receive a balanced reward for their efforts.

## 1. Introduction

Coronaviruses have caused a series of pandemics: the severe acute respiratory syndrome (SARS) in 2002, Middle East respiratory syndrome (MERS) in 2012, and coronavirus disease (COVID-19) in 2020. Three years and four months after the first case was reported in Wuhan, China, the World Health Organization (WHO) declared the COVID-19 pandemic over. Although vaccine development has significantly controlled COVID-19 infections, projections of future scenarios suggest the possibility of years of coexistence with COVID-19, emphasizing the need to address the psychological aftermath of the pandemic [1,2].

In Korea, the number of clinical nurses in healthcare organizations in 2020 was 7.9 per 1000 individuals, which is lower than the Organization for Economic Co-operation and Development (OECD)’s average of 9.4. A study of 243,077 nurses registered with the Korea Health Insurance Review and Assessment Service showed that the turnover rate, mainly among nurses aged below 30 years, was 48.2%; it was 57% among nurses with less than one year of experience; and the average tenure of the nurses who left was 2.3 years [3]. The COVID-19 pandemic posed a significant threat to nurses working in hospitals because of the urgency and uncertainty of the situation. Additionally, nurses had to cope with the hospitals’ changed infection control policies and procedures, and excessive and high-intensity work. This was reflected in the fact that among healthcare workers, nurses have the highest turnover intentions at 3.79 out of 5 [4]. Notably, 10.6% of the nurses who directly cared for patients of COVID-19 in South Korea considered changing jobs during the pandemic [5].

A meta-analysis during the COVID-19 pandemic found that healthcare workers experienced burnout (37.4%), anxiety (34.4%), depression (31.8%), stress (40.3%), post-traumatic stress syndrome (11.4%), insomnia (27.8%), and psychological distress (46.1%) [6]. A total of 51.4% of healthcare workers who had direct contact with patients of COVID-19 experienced burnout [7], with emotional exhaustion being particularly high among the three sub-domains of burnout. Post-traumatic stress disorder (PTSD) was reported by 11–74% of healthcare workers, with 10–40% of subjects developing symptoms 1–3 years later [8]. It was also associated with increased turnover intentions [9,10].

After the SARS outbreak in Canada, frontline healthcare workers reported higher levels of burnout, psychological distress, and PTSD compared to those who did not care for patients with SARS [11]. Nurses’ own experience of caring for patients of COVID-19 was a major contributing factor of their poor mental health [12]. Additionally, nurses’ mental health is a significant predictor of turnover intentions [13].

The effort–reward imbalance (ERI) model [14] is among the main theories used to explain job stress. An effort–reward imbalance implies a lack of reciprocity between job-related effort and rewards such as pay, respect, and job security, resulting in a situation of excessive effort and low rewards [15]. This situation elicits strong negative emotions, has long-term adverse effects on health, and triggers a persistent stress response. Conversely, appropriate rewards can promote positive emotions and well-being [16]. Overcommitment, included in the ERI model, is a personality construct that refers to an individual’s need for control in dealing with the demands of work. Individuals with high overcommitment show patterns of attitudes, behaviors, and emotions characterized by excessive effort on tasks and strong motivation for respect and approval at work [16]. The effort–reward ratio refers to the ratio of the extrinsic effort score to the reward score, and a score greater or less than one indicates an effort–reward imbalance. Effort–reward imbalance among hospital nurses is a significant influencing factor of their turnover intention [17].

Recently, rewards have emerged as a significant mechanism to reduce turnover intention. Rewards include not only monetary benefits in return for job efforts based on social exchange relationships, but also job security, workplace respect, and the expectation of promotion [18].

People in helping professions may experience positive feelings of compassion satisfaction by helping those for whom they feel compassion. However, they may also experience negative feelings of compassion fatigue from knowing that the person they are helping has experienced a traumatic event [19]. Compassion satisfaction refers to the benefits that caregiving professionals derive from working with people in distress [20]. It is a source of strength for helping professionals to continue working in the face of high risk or high stress, and is a factor that lowers burnout and improves quality of life [21]. The opposite concept, compassion fatigue, refers to a state of dysfunction and burnout due to exposure to compassion-induced stress [22]. It is categorized as both burnout and secondary traumatic stress [19].

Using the concept of professional quality of life, Stamm [19] introduced the concept of compassion satisfaction as a protective factor against the negative effects of compassion fatigue and burnout in professional care providers after experiencing a traumatic event. Another study of tertiary hospital nurses revealed that those with lower levels of compassion satisfaction had significantly higher turnover intentions than those with higher levels of compassion satisfaction [23,24].

Burnout is the result of a psychological response to chronic work stress, and the psychological symptoms are also expressed in physical symptoms [25]. Generally, nurses are more susceptible to burnout compared to other healthcare workers due to the nature of their profession, which involves working in close proximity to patients and caring for them for long periods of time [26]. Symptoms of burnout include physical, emotional, and psychological symptoms such as frustration, fatigue, anxiety, worry, depression, insomnia, lethargy, loss of motivation, skepticism, self-hatred, and negative self-concept due to extreme psychological and physical exhaustion, which manifest as personal problems [27]. These personal problems negatively impact family members, colleagues, business partners, and customers at the interpersonal level. At the organizational level, they pose serious problems such as low job satisfaction, decreased organizational commitment, absenteeism, turnover, resignation, and dishonest work attitudes [28]. Detecting burnout syndrome is difficult because it begins without any preceding symptoms; the process remains unrecognized until a certain point, after which the symptoms suddenly appear. It is like a disease from which recovery is difficult because once the symptoms appear, they are progressive and continuous and can rapidly metastasize [29].

Nursing is among the human service professions that are particularly vulnerable to burnout syndrome globally [30]. Burnout is characterized by three typical symptoms: exhaustion, depersonalization, and reduced personal accomplishment [31]. Once high burnout occurs, turnover is likely to occur [32].

According to the fifth edition of the Diagnostic and Statistical Manual of Mental Disorders (DSM-5), trauma refers to direct trauma and vicarious trauma. The former includes experiencing or feeling actual or threatened death, serious injury, or sexual violence. The latter includes learning that a traumatic event has occurred to a close family member or close friend or being repeatedly exposed to a traumatic event [33].

Calhoun and Tedeschi defined trauma as the experience of a threatening event that disrupts one’s existing schemata, values, and beliefs about the world. It is so threatening that adaptation via existing cognitive schemata or coping mechanisms is not possible. The researchers emphasized the subjective rather than the objective perception of threat [34].

Traumatic stress is not limited to those who have been directly victimized. It also affects those who have witnessed the traumatization of others or have family or friends who have been directly victimized [22], as well as those who have indirectly experienced a traumatic event, such as practitioners working with victims [22,35].

Particularly, the behavioral and emotional changes that occur in the helper during the process of helping a traumatized person, and that exhibit the same symptoms of post-traumatic stress as the symptoms of intrusion, avoidance, and hyperarousal, are referred to as secondary traumatic stress [22]. Nurses with secondary traumatic stress may experience challenges with problem-solving and decision-making, difficulty concentrating, sleep disturbances, intrusive thoughts, anxiety, fear of the future, and decreased activity levels. Chronic exposure to secondary traumatic stress can lead to job dissatisfaction, chronic illness, and interpersonal problems, as well as adversely impact clinical performance and the quality of patient care [22].

Secondary traumatic stress is influenced by the experience and intensity of the traumatic event, external factors such as duration of exposure, and the internal and external resources of the individual [36]. A key internal resource that influences secondary traumatic stress is resilience, which is a personal characteristic that enables a person to thrive in the face of adversity [37] and is considered a tool for coping with stress. Particularly, resilience acts as a counterweight when a person experiences a traumatic event that may upend their life, moderating the progression of traumatic event stress, job stress, etc., to PTSD, mental health, and behavioral problems [38,39]. Additionally, low resilience in nurses can lead to burnout, compassion fatigue, and chronic illness, and these negative outcomes are similar to those that arise from secondary traumatic stress [37].

A study of nurses caring for patients of COVID-19 in South Korea found that secondary traumatic stress averaged 31.23 to 43.23 out of a total score of 50 [40,41]. A study of 2153 healthcare providers in India revealed that 77% of the subjects experienced secondary traumatic stress, with 8.2% of them experiencing severe levels [42].

A systematic review of factors related to turnover intention among Korean hospital nurses over a 10-year period up to 2016 [13] identified that various influencing factors have already been studied and significant factors have been identified, including personal factors, family and job factors, and ward factors. However, there are few studies on factors such as mental health, professional quality of life, effort–reward imbalance, etc.

Therefore, this study aims to provide a basis for more efficient human resource management by identifying the effects of mental health, professional quality of life, and the effort–reward imbalance on turnover intention among hospital nurses who worked during the third year of the COVID-19 pandemic.

The specific objectives include:Determining the extent of turnover intention according to participants’ general and clinical characteristics;Understanding the relationship between mental health, compassion satisfaction, burnout, secondary traumatic stress, effort–reward imbalance, and turnover intentions;Identifying correlations among mental health, compassion satisfaction, burnout, secondary traumatic stress, effort–reward imbalance, and turnover intention;Identifying factors influencing employee turnover intentions.

## 2. Materials and Methods

### 2.1. Study Design and Participants

This descriptive survey explores the factors influencing nurses’ turnover intention during the COVID-19 pandemic. The participants included 250 nurses working at the C University Hospital in the Gyeonggi Province for more than one year during the COVID-19 pandemic, and working in 13 wards, 5 ICUs, and 2 emergency centers, excluding special parts, operating rooms, and outpatient departments less related to COVID-19.

The sample size was calculated using G*Power 3.1.9.7. Setting a medium effect size of 0.15, two-tails, significance level (α) of 0.05, and statistical power (1-β) of 0.95 for the regression analysis, 222 cases were calculated as necessary for the study with 20 predictors, including patient characteristics and study variables. A final sample size of 250 cases was targeted with an expected dropout rate of approximately 10%.

### 2.2. Measures

#### 2.2.1. Psychological Health

The General Health Questionnaire estimates the likelihood of mental illness and is widely used worldwide as a mental health measure for non-psychiatric healthcare professionals. This study used the Korean General Health Questionnaire (KGHQ-20) [43], which was standardized in Korean. There are several versions of this tool depending on the number of questions. This study used the KGHQ-20, which comprises depression (five questions), anxiety (four questions), social dysfunction (10 questions), and frequency of going out (one question).

Respondents had to compare their psychological state over the past two to three weeks with their usual psychological state to identify problems in their current state. All items were scored on a 4-point Likert scale, with 3 being very much so and 0 being not at all so; higher scores indicated higher levels of psychological distress. Higher sum scores represented higher levels of psychological distress. The cut-off scores using the GHQ Likert format were 23/24 [44] or 27/28 [45,46]. At the time of development [43], Cronbach’s alpha was 0.87, and in this study, it was 0.875.

#### 2.2.2. Effort–Reward and Overcommitment

Effort–reward imbalance was measured using the ERI questionnaire developed by Siegrist et al. [15] and standardized into Korean by Eum K.D. [47]. The items for effort and reward included 5 questions for “effort”, 11 for “reward”, and 6 for “overcommitment”.

The items for effort and reward included five response categories and were answered in two steps. First, the respondents agreed or disagreed with whether the item content described a typical experience of their workplace. If they agreed, they were asked to evaluate to what extent they usually felt distressed. Each item of “effort” and “reward” was scored on a 5-point scale. Thus, a total sum score based on the five items measuring “effort” varied between 6 and 30—the higher the score, the more the perceived demands were experienced as stressful. The total score for “reward” varied between 11 and 55. A score of 11 indicated the perception of the lowest reward; 55 reflected a very high level of reward (reverse coding).

The effort–reward ratio was computed for every respondent according to the formula: e/(r × c), where “e” represented the sum score of the effort scale, “r” was the sum score of the reward scale, and “c” defined a correction factor for different numbers of items in the numerator and denominator. The correction factor to adjust for the number of items was 0.54545 (=6/11).

To determine the degree of effort–reward imbalance, the effort–reward ratio was calculated by dividing the total score of the effort domain by the total score of the reward domain [18]. This value was calculated by multiplying the total score of the reward section by 6/11 (=0.5454) as the denominator and the total score of the effort section as the numerator [(effort)/(reward) × 0.5454]. This was performed because the number of questions in the effort and reward sections were 6 and 11, respectively.

In this case, if the value of the effort–reward ratio exceeded 1, it implied that the reward was low compared to the effort. If it was less than 1, it implied that the reward was high compared to the effort. Consequently, a value close to zero indicated a favorable condition, relatively low effort, and relatively high reward. Values beyond 1.0 indicate that a lot of effort was expended relative to the reward.

The overcommitment items were scored on a 4-point Likert scale ranging from 4 (strongly agree) to 1 (strongly disagree) and computed to a total score varying from 6 to 24. The higher the score, the more likely a subject was to experience overcommitment at work.

The reliability of the tool was based on previous research [47], where Cronbach’s alpha values were 0.71 for effort, 0.86 for reward, and 0.75 for overcommitment. In this study, they were 0.795, 0.785, and 0.795, respectively.

#### 2.2.3. Professional Quality of Life

Professional quality of life refers to the subjective evaluation of the quality of life felt by individuals who help others in their profession. It comprises both positive and negative aspects related to job performance. The positive aspect includes compassion satisfaction and the negative aspect consists of compassion fatigue, which is further divided into burnout and secondary traumatic stress [19].

Compassion satisfaction is a positive aspect of the impact that professional caregivers may experience while caring for diverse populations [19]. Burnout is a negative health syndrome characterized by an individual’s physical, sexual, or emotional exhaustion in response to a source of stress [19]. Secondary traumatic stress refers to the negative emotional consequences that professional caregivers experience as a result of caring for clients who have been physically or emotionally injured [19].

This study measured compassion satisfaction and compassion fatigue using the Korean version of the Professional Quality of Life (ProQoL) compassion satisfaction/fatigue sub-scale version 5.0, which was developed by Figley [22] and modified by Stamm [19]. The instrument includes 30 items that are rated on a 5-point Likert scale; there are 10 items each for compassion satisfaction, burnout, and secondary traumatic stress. The scale ranges from 1 (not at all) to 5 (very much so); higher scores indicate higher levels of compassion satisfaction, burnout, and secondary traumatic stress. For each variable, a total score of 22 or less indicates “low”, 23–41 indicates “moderate”, and 42 or more indicates “high”. In Stamm’s study [19], Cronbach’s alpha values for compassion satisfaction, burnout, and secondary traumatic stress at the time of development were 0.88, 0.81, and 0.75, respectively. In this study, they were 0.927, 0.726, and 0.832, respectively.

#### 2.2.4. Turnover Intention

Turnover intention refers to an organizational member’s intention to voluntarily leave their current job in the near future and is measured using an instrument developed by Lawler [48]. This instrument was modified for nursing organizations by Park [49]. The instrument comprises four items, each measured on a 5-point Likert scale ranging from 1 (not at all true) to 5 (very true); higher scores indicate a higher turnover intention. The reliability of the instrument in Park’s study [49] was represented by a Cronbach’s alpha of 0.88 [49], while it was 0.885 in this study.

### 2.3. Data Collection

This study was conducted after obtaining permission from the nursing department of the C University Hospital in the Gyeonggi Province, South Korea, and after review and approval by the Institutional Review Board of the C University Hospital. A recruitment notice was posted in front of the hospital’s outpatient reception area. After seeking cooperation from the head nurse of each ward, nurses in the department were asked to participate voluntarily, and written participation consent was obtained. The average time spent by the respondents was less than 20 min. They were informed that they could stop at any time during the survey. The data were collected in January 2023.

### 2.4. Statistical Analysis

The collected data were analyzed using the IBM SPSS 25 statistical program. General characteristics, clinical characteristics, and research variables were analyzed using descriptive statistics, and comparisons of turnover intention based on general characteristics were analyzed using the *t*-test and ANOVA, and post hoc analysis was performed using Scheffe. Pearson’s correlation coefficient was used to determine the relationship between each variable, and the hierarchical multiple regression analysis was applied to analyze the influencing factors of the subjects’ turnover intention.

### 2.5. Ethical Consideration

This study was conducted after review and approval by the Institutional Review Board of the C University Hospital (UC22QISI0095). The consent form clarified the purpose of the study, participation period, research procedures, rights of the subjects, risks or inconveniences, privacy, voluntary consent, the possibility of withdrawal from the study without penalty, and that the data would not be used for any purpose other than the purpose of the study. The subjects who completed the study were rewarded with coffee coupons worth KRW 5000.

## 3. Results

### 3.1. Participants’ Characteristics and Differences in Turnover Intention

The total number of participants in this study was 248: 93.5% (232) females, 6.5% (16) males; mean age 31.6 ± 6.4 years, 47.6% (118) were in their 20s; 61.7% (153) were single; 54.8% (136) had no religion; and 58.5% (145) were nurses with a four-year bachelor’s degree. In terms of the department, 33.9% (84) were ICU nurses and 68.1% (169) were general nurses, with a mean total nursing experience of 8.7 ± 6.6 years, of which 33.1% (82) had 5–10 years (61–120 months) of experience. The average work experience in the current hospital was 8.3 ± 6.7 years, with 26.2% (65) having 3–5 years (37–60 months) of work experience. The average number of hours worked per day was 8.6 ± 0.8, with 54.8% (136) nurses working 8 h or less. A total of 18.5% (46) had changed jobs, 37.9% (94) had worked in an isolation room during the COVID-19 pandemic, and 58.1% (144) had worn personal protective equipment (PPE). A total of 0.8% (two nurses) had ever been infected with COVID-19 (Table 1).

The differences in the turnover intention based on the subjects’ general characteristics were as follows. In terms of sex, the turnover intention of women was significantly higher (t = −2.426, *p* = 0.025), and in terms of age, those in their 20s and 30s had a higher turnover intention than those in their 40s (F = 6.890, *p* = 0.001). In addition, the turnover intention was higher for the unmarried group (t = 2.584, *p* = 0.010), the no religion group (t = −3.225, *p* = 0.001), and those with associate’s and bachelor’s degrees than those with master’s degrees (F = 6.418, *p* = 0.002).

When examining differences in the turnover intention based on clinical characteristics, the work unit was higher in the general wards, intensive care units (ICUs), emergency departments, and comprehensive nursing care service (CNCS) wards (F = 3.229, *p* = 0.023), but post hoc tests showed no difference between groups. The position was significantly higher among nurses and seniors (F = 9.853, *p* = <0.001). There were significant differences in the turnover intention based on total work experience (F = 3.842, *p* = 0.001) and current hospital experience (F = 3.867, *p* = 0.001), but post hoc tests showed no differences between groups.

Otherwise, there were no statistically significant differences in turnover intention based on the number of hours worked per day, previous job change, experience of self-isolation during the COVID-19 pandemic, experience of working in personal protective equipment (PPE) during the pandemic, or experience of COVID-19 infection (Table 1).

### 3.2. Descriptive Statistics of the Research Variables

The main concepts of the subjects and the means for each sub-domain are as follows. Psychological distress averaged 24.3 ± 6.8 points, and when compared according to cut-off scores, 10/11 had 14.1% (35 people) with 11 or more, 23/24 had 54.4% (135 people) with 24 or more, and 27/28 had 30.6% (76 people) with 28 or more. When comparing turnover intention, the mean of cut-off score 11 or above (16.0 ± 3.5), 28 or above (15.3 ± 3.8), and 24 or above (14.0 ± 4.0) were higher.

The mean of effort, reward, and overcommitment were 15.0 ± 5.2, 49.2 ± 5.8, and 14.8 ± 2.9, respectively. Only 0.8% (2 people) had an effort–reward ratio of 1.00, and 21.2% (53 people) had an effort–reward ratio of 1.01 or higher, implying that effort was higher than reward. The turnover intention scores averaged 15.6 ± 3.5, which was significantly higher than 1.00.

Among the professional quality of life sub-factors, compassion satisfaction averaged 29.5 ± 7.8, of which 18.1% (45 people) were in the low compassion satisfaction group (22 points or less). Their turnover intention scores averaged 16.3 ± 3.0, which was significantly high. The secondary traumatic stress averaged 22.8 ± 6.5, of which 45.2% (112 people) scored moderate or higher (23–41 points), with a significantly higher mean of 13.8 ± 3.8 compared to turnover intention. The burnout scores averaged 28.3 ± 5.6, with 1.6% (4 people) in the high burnout group (42 points or more) having significantly higher turnover intentions, averaging 18.5 ± 3.0. The average turnover intention score for this group was 13.2 ± 4.0 (Table 2).

### 3.3. Correlation among the Research Variables

When examining the correlations among the variables and turnover intention, burnout (r = 0.578, *p* < 0.001), effort (r = 0.390, *p* < 0.001), overcommitment (r = 0.354, *p* < 0.001), and psychological distress (r = 0.345, *p* < 0.001) had significant positive correlations with the turnover intention. Compassion satisfaction (r = −0.530, *p* < 0.001) and compensation (r = −0.470, *p* < 0.001) had significant negative correlations (Table 3).

### 3.4. Factors Affecting Participants’ Turnover Intention

To conduct the regression analysis, the autocorrelation of the dependent variable and the multicollinearity between the independent variables were examined. The autocorrelation of the dependent variable was calculated using the Durbin–Watson index, which was 2.049, indicating that it was independent without autocorrelation. Multicollinearity among independent variables was examined using the VIF index, which ranged from 1.101 to 6.530 and was less than 10, indicating no multicollinearity. Therefore, the data were suitable for regression analysis.

To examine the effect of the study variables on turnover intention while controlling for sociodemographic and clinical characteristics, a hierarchical regression analysis was conducted in which sociodemographic and clinical characteristics were entered in Step 1 and the study variables in Step 2. In Model 1, which included sociodemographic and clinical characteristics as control variables, 13.3% of the variance was explained by general ward (β = −0.197, *p* = 0.004), emergency department (β = −0.188, *p* = 0.011), seniority (β = 0.285, *p* = 0.019), and the presence of protective clothing (β = 0.141, *p* = 0.022).

In Model 2 with the independent variables, among the sociodemographic and clinical characteristics, the general ward (β = −0.112, *p* = 0.049) had a significant effect on the turnover intention. Among the independent variables, compassion satisfaction (β = −0.349, *p* = <0.001), burnout (β = 0.224, *p* = 0.015), effort–reward ratio (β = 0.158, *p* = 0.014), and psychological distress GHQ 23/24 (β = 0.123, *p* = 0.036) were significant, with 44.1% explanatory power (Table 4).

## 4. Discussion

This study aimed to determine the mental health, effort–reward and overcommitment, professional quality of life, and turnover intention of hospital nurses during the COVID-19 pandemic. Additionally, it attempted to identify the influencing factors of nurses’ turnover intention.

It compared the level of turnover intention based on general and clinical characteristics, and identified that women in their 20s and 30s, those who were single, those who were non-religious, and diploma and bachelor’s degree holders had higher turnover intention. Additionally, those working in general wards and ICUs, and those with 13 months to 15 years of clinical experience tended to have higher turnover intention. In previous studies, age 20–30 [50,51,52], being single [13,50], being non-religious [13], diploma and bachelor’s degree holder [53], having clinical experience in order of 1–5 years, 5–10 years, and 10–15 years [52], general nurse and charge nurse [50], which supported the findings of this study as factors that increase turnover intention. There was no difference in turnover intention between nurses in general wards and nursing care integrated service wards [54], and the turnover intention of nurses in nursing care integrated service wards was relatively higher than the results of this study [55]. In this study, the significantly lower turnover intentions of subjects with higher age and experience, religion, marriage, and master’s degree or higher compared to other subjects may be due to the fact that they have the conditions to maintain a stable hospital life. The higher degree of turnover intention among nurses in general wards is thought to be due to the fact that nursing care integrated service wards prefer to admit mild patients in consideration of patient safety such as falls, and general wards have a higher work intensity by admitting mainly critically ill patients who require guardianship. In the case of intensive care units, due to COVID-19, guardian visits are completely restricted and only 1:1 telephone interviews are possible, which is thought to have increased the intensity of work due to the additional workload. In the event of a new infectious disease outbreak in the future, it is necessary to supplement the system for nurses’ patient safety and guardian response so that their work performance can be stabilized. In this study, nurses’ turnover intention averaged 3.3 out of 5, which varied from 3.18 to 3.89 among general hospital nurses using the same instrument [17,52,53,55,56,57,58,59,60]. 

In a meta-analysis [13], the combined turnover intention score was 3.26, indicating that nursing is a profession with a higher than moderate turnover intention. Due to the high turnover rate of nurses, there are currently only 193,900 active nurses out of a total of 395,000 licensed nurses in South Korea, or 49.1% of the total [61]. The wide variation in turnover intention scores may be due to different influencing factors, which should be analyzed. Reducing turnover intention may be a strategy for reducing actual turnover. 

Nurses’ effort–reward was measured using a 5-point scale, and a 4-point scale was used to measure overcommitment. A study of workers in a petrochemical company [47] showed the following: total mean effort 15.0 (item mean 2.5 out of 5), reward 36.8 (item mean 3.3 out of 5), and overcommitment 14.8 (item mean 2.5 out of 4). This study showed lower effort and overcommitment (11.72 [1.95 points] and 13.58 [2.26 points], respectively), and higher reward 47.96 (4.36 points). In a study of Ecuadorian nurses [62], effort was 11.17 and reward was 22.4—lower than the results of this study. Additionally, the average effort–reward ratio in this study was 0.8, which was similar to the range of 0.57 to 0.82 in seven European countries [63]. We also found that 21.2% (53 people) of our subjects had an effort-reward raio of 1.01 or higher, which is similar to the 22.8% in Germany, which scored the highest among the seven European countries. In another study [17], the effort–reward ratio was greater than 1; that is, 86.5% of the nurses perceived they were investing more effort, compared to 21.2% in this study, indicating that effort is less rewarded. A study of police officers [64] revealed an effort–reward ratio of 1.32, indicating high effort compared to reward, and an overcommitment level of 2.90, which was higher than that of the subjects in this study. Although we cannot make a direct comparison of the effort–reward ratio for nurses owing to the lack of similar studies, we can see that the nurses in this study are relatively satisfied with the effort–reward ratio and achieving a positive effort–reward ratio, because the ratio is high. To efficiently manage human resources in the future, national policies, remuneration in medical institutions, and the welfare of nurses should be examined so that the effort–reward ratio can be kept below one.

The professional quality of life of nurses in this study comprises compassion satisfaction (29.5 points), secondary traumatic stress (22.8 points), and burnout (28.3 points), and we would like to compare them with the results of previous studies. According to Stamm’s [19] categorization and semantic partial scores, the scores were in the moderate level between 23 and 41, and only secondary traumatic stress was borderline. Specifically, previous studies that applied the same instrument to nurses [65,66,67,68] identified that the scores ranged from 26.5 to 34.0, secondary traumatic stress from 25.2 to 32.1, and burnout from 28.0 to 30.7. Overall, professional quality of life scores tended to be higher in studies conducted after 2019, that is, during the COVID-19 pandemic [67,68].

Conversely, a previous study of Canadian hospice nurses [69] found that compassion satisfaction was significantly higher and secondary traumatic stress and burnout were lower in Korean hospice nurses, with compassion satisfaction at 43.4, secondary traumatic stress at 20.1, and burnout at 22.3. The Korean hospice nurses had higher compassion satisfaction and secondary traumatic stress scores of 34.41 and 27.38, respectively, compared to the participants in this study. Hospice nurses have more time to build empathy with patients in the hospice work environment [70]. Additionally, they have more opportunities to share information about their personal lives in addition to physical care, which is interpreted as a higher compassion satisfaction in providing care to their patients [71].

Thus, the degree of professional quality of life varies by country and by nurses’ field of work. The fact that domestic studies showed similar results suggests that empathic nursing care is provided to humans in the same cultural environment, and although there are always many variables in practice, professional quality of life may be similar. Therefore, it is necessary to formulate policies to enhance compassion satisfaction and reduce secondary traumatic stress and burnout to improve the quality of nursing work.

In this study, turnover intention was positively related to burnout, effort, overcommitment, and mental health, and significantly negatively related to compassion satisfaction and compensation. In previous studies, burnout [55], effort–reward imbalance [17], psychosocial distress [72], and compassion satisfaction [24] were significantly related to turnover intention, and compassion satisfaction was negatively related to turnover intention, supporting the results of this study. In other studies, mental health [73] was significantly related to turnover intention, but there was no significant difference in the influencing factors. Shkembi et al. [74] found that over-engagement was significantly related to poor mental health, and Lee Jung-hoon [51] identified in a study of critical care nurses that they thought of leaving because of dissatisfaction with compensation, absence of break time guarantee, and lack of hope that the work environment would improve compared to high work intensity, supporting the result that the lower the compensation, the higher the turnover intention. Therefore, it is necessary to reduce job demands and increase job resources for compassion satisfaction and compensation so that burnout, effort, overcommitment, and psychological distress can be reduced.

In this study, in Model 1, which included general and clinical characteristics as control variables, the following were identified as the influential factors of turnover intention: general ward, emergency room, pre-promotion general nurses and first-promotion nurses, and working with protective clothing. Additionally, working in general wards and emergency departments had a significant effect on nurses’ turnover intention. This supports the results of previous studies [52,53] that found that working in general wards had a significant effect on turnover intention. A previous study [75] investigated the turnover of nurses in senior general hospitals from 2009 to 2013 and identified that the turnover rate of nurses was 9.9% in general wards, 8% in operating rooms and emergency rooms, and 13.1% in intensive care units. In other words, it is speculated that nurses working in intensive care units that require parental presence have high turnover intentions. Therefore, it is necessary to control turnover intention by dispersing critical patients instead of concentrating them in general wards or implementing nursing care in full. Additionally, it is crucial to improve the quality of emergency department operations and provide additional compensation to emergency department workers, such as incentives for evaluating nursing care integrated wards.

A previous study [76] found that general nurses had a higher turnover intention than chief manager nurses and above, and the higher the position, the lower the turnover intention. This result is supported by the fact that the higher the position, the more familiar the work, and the higher the compensation, the relatively better the job satisfaction, which lowers the turnover intention [77]. The absence of promotion experience had a significant effect on turnover intention [78]. As nurses gain more seniority, their burden to work with junior nurses increases, which can lead to increased turnover intentions if promotions are not forthcoming. Therefore, it is necessary to improve wages and promotion prospects as a way to recognize nurses’ work and experience. This study assumed that the above chief manager nurses had attained economic and social stability, and therefore, their turnover intention was low.

In this study, the presence or absence of working in protective clothing had a significant impact on turnover intention. A 2021 study [79] reported that the longer the duration of wearing protective clothing, the higher the turnover intention. However, it was not statistically significant because that was a time of chaotic work environment, and wearing protective clothing alone did not increase turnover intention. As this study was conducted three years after the outbreak of the COVID-19 pandemic, it was assumed that the continuous wearing of protective clothing significantly impacted the turnover intention.

In Model 2, which includes the independent variables, compassion satisfaction, burnout, effort–reward ratio, and mental distress were found to significantly impact the order of the independent variables. In this study, the higher the compassion satisfaction in professional quality of life, the lower the turnover intention, and the higher the burnout, the higher the turnover intention. In a previous study [24] compassion satisfaction was divided into the low, moderate, and high groups depending on the score. When comparing turnover intention and non-turnover intention, the turnover intention group showed a significantly lower level of compassion satisfaction compared to the non-turnover intention group. In particular, the percentage of high compassion satisfaction in the turnover intention group was only 1%, whereas the percentage of low compassion satisfaction was relatively high at 8.7%. Compassion satisfaction, as opposed to compassion fatigue, could facilitate the continuation of the performance of helpful work even in highly dangerous or stressful work situations, which can reduce compassion fatigue and burnout [21]. Caregivers reported feeling compassion satisfaction upon the client’s recovery and return to their daily activities; when supported by supervisors and coworkers; and when feeling a strong sense of camaraderie within their work environment. Additionally, this compassion satisfaction also improved the healthcare worker’s self-esteem, making them more confident in their helping behaviors [80]. As compassion satisfaction acts as a moderating factor in reducing burnout, individual efforts and organizational support are crucial for nurses to feel positive emotions [81]. This study is significant because compassion satisfaction is a factor that significantly affects turnover intention. Therefore, it is necessary to repeat the study in the future because there are few studies on the relationship between compassion satisfaction and turnover intention [24]. Compared to physicians, nurses showed higher scores in compassion satisfaction [67], and compassion satisfaction was found to be a major factor affecting burnout. Burnout is highly negatively related to turnover intention and has been the focus of burnout research [81]. Based on these results, it is necessary to promote positive factors such as compassion satisfaction and reduce negative factors such as burnout to reduce turnover intention.

## 5. Conclusions

This study was conducted to determine the mental health, effort–reward and overcommitment, professional quality of life, and turnover intentions of hospital nurses during the COVID-19 pandemic and identify the influencing factors of nurses’ turnover intentions. The results showed that the main factors affecting turnover intention were compassion satisfaction, burnout, effort–reward ratio, and mental distress. The higher the compassion satisfaction, the lower the turnover intention, and the higher the burnout, effort–reward imbalance, and mental distress, the higher the turnover intention.

Therefore, to reduce nurses’ turnover intention, organizations should aim to improve compassion satisfaction. Additionally, institutional and national policies should be designed to minimize nurses’ effort–reward imbalance. Interventions and policies for burnout and mental distress are also urgently required.

### Limitations and Future Research Directions

First, this study collected data via convenience sampling from nurses working in hospitals in one region. Future studies should include a larger sample from multiple medical institutions from multiple regions. Second, as there is no mental health (GHQ-20) measurement tool for nurses, it is necessary to repeat this study in the future. Third, there is a lack of research on nurses’ effort–reward imbalance. Therefore, future research should identify the extent of effort–reward imbalance among hospital nurses and the influencing factors. Fourth, we recommend a study to identify the relationship between mental health (GHQ-20) and effort–reward imbalance and nurse turnover.

## Figures and Tables

**Table 1 healthcare-11-02695-t001:** Differences in turnover intention according to participants’ characteristics.

Characteristics	Categories	n	%	M ± SD	t or F	*p*	Scheffé
gender	female	232	93.5	13.3 ± 4.0	−2.426	0.025	
	male	16	6.5	11.5 ± 2.7			
age M ± SD = 31.6 ± 6.4	≤29 ^a^	118	47.6	13.6 ± 3.6	6.890	0.001	c < a = b
	30~39 ^b^	97	39.1	13.4 ± 4.0			
	≥40 ^c^	33	13.3	10.8 ± 4.4			
marital statue	single	153	61.7	13.7 ± 3.8	2.584	0.010	
	married	95	38.3	12.3 ± 4.1			
religion	yes	112	45.2	12.3 ± 4.2	−3.225	0.001	
	no	136	54.8	13.9 ± 3.6			
education status	3 year college ^a^	60	24.2	13.9 ± 3.6	6.418	0.002	c < a = b
	4 year college ^b^	145	58.5	13.4 ± 3.8			
	graduate above ^c^	43	17.3	11.3 ± 4.5			
type of department	general ward	36	14.5	11.9 ± 4.0	3.229	0.023	
	CNCS wards *	69	27.8	13.9 ± 4.4			
	ICU	84	33.9	13.6 ± 3.6			
	ER	59	23.8	12.4 ± 3.7			
job position	staff nurse ^a^	169	68.1	13.5 ± 3.6	9.853	<0.001	c < a = b
	assistant manager nurse ^b^	46	18.5	14.0 ± 4.2			
	above chief manager nurse ^c^	33	13.3	10.5 ± 4.1			
total clinical experiences (month)	≤12	6	2.4	10.5 ± 3.0	3.842	0.001	
	13~36	36	14.5	13.1 ± 2.8			
	37~60	48	19.4	13.9 ± 3.7			
	61~120	82	33.1	13.9 ± 4.1			
	121~180	36	14.5	13.6 ± 3.9			
	181~240	22	8.9	11.2 ± 4.7			
	≥241	18	7.3	10.4 ± 3.5			
total experiences in present hospital (month) M ± SD = 8.3 ± 6.7	≤12	9	3.6	11.0 ± 2.5	3.867	0.001	
	13~36	38	15.3	13.2 ± 3.0			
	37~60	65	26.2	14.0 ± 3.6			
	61~120	63	25.4	13.7 ± 4.3			
	121~180	34	13.7	13.7 ± 3.9			
	181~240	21	8.5	11.1 ± 4.8			
	≥241	18	7.3	10.4 ± 3.5			
work hours per day M ± SD = 8.6 ± 0.8	≤8	136	54.8	12.9 ± 3.8	−1.356	0.176	
	≥9	112	45.2	13.5 ± 4.1			
turnover experience	yes	46	18.5	13.5 ± 3.8	−0.560	0.576	
	no	202	81.5	13.1 ± 4.0			
experience of self-isolation during COVID-19	yes	94	37.9	13.4 ± 4.2	−0.822	0.412	
	no	154	62.1	13.0 ± 3.8			
experience of wearing PPE ** during COVID-19	yes	144	58.1	13.6 ± 4.1	−1.891	0.060	
	no	104	419	12.6 ± 3.7			
COVID-19 infection history	yes	2	0.8	13.0 ± 2.8	0.058	0.950	
	no	246	99.2	13.2 ± 4.0			

*N* = 248, * CNCS wards: comprehensive nursing care service wards, ** PPE: personal protective equipment. ICU = Intensive Care Unit; ER = Emergency Room.

**Table 2 healthcare-11-02695-t002:** Level of psychological distress, professional quality of life, effort–reward, and overcommitment and turnover intention.

Variables			n(%)	min	max	M ± SD	Item M ± SD	Turnover Intention	
								M ± SD	t or F (p)
**psychological distress total**				5	42	24.3 ± 6.8	1.2 ± 0.3	-	-
anxiety				5	17	11.6 ± 2.2	2.3 ± 0.4	-	-
depression				4	14	8.9 ± 1.9	2.2 ± 0.5	-	-
social maladjustment				11	31	21.6 ± 3.2	2.2 ± 0.3	-	-
cut-off score	10/11 *	≤10	213(85.9)	0	10	4.6 ± 3.0	-	12.7 ± 3.8	−4.783 (<0.001)
		≥11	35(14.1)	11	17	13.1 ± 1.7	-	16.0 ± 3.5	
	23/24 **	≤23	113(45.6)	5	23	18.4 ± 4.3	0.9 ± 0.2	12.2 ± 3.7	3.573 (<0.001)
		≥24	135(54.4)	24	42	29.2 ± 4.2	1.5 ± 0.2	14.0 ± 4.0	
	27/28 **	≤27	172(69.4)	5	27	20.9 ± 4.9	1.0 ± 0.2	12.2 ± 3.6	−5.982 (<0.001)
		≥28	76(30.6)	28	42	32.1 ± 3.5	1.6 ± 0.2	15.3 ± 3.8	
**effort-reward imbalance**								-	-
effort				6	30	15.0 ± 5.2	2.5 ± 0.9	-	-
reward				25	55	49.2 ± 5.8	4.5 ± 0.5	-	-
overcommitment				7	23	14.8 ± 2.9	2.5 ± 0.6	-	-
effort-reward ratio ***		total		0.27	1.90	0.8 ± 0.3	-	-	-
		<1.00	193(78.0)	0.27	0.99	0.6 ± 0.2	-	12.5 ± 3.8	−5.349 (<0.001)
		=1	2(0.8)	1.00	1.00	1.0 ± 0.0	-		
		≥1.01	53(21.2)	1.01	1.90	1.2 ± 0.2	-	15.6 ± 3.5	
**professional quality of life**								-	-
compassion satisfaction	total			10	50	29.5 ± 7.8	2.9 ± 0.8	-	-
	low (≤22) ^a^		45(18.1)	10	22	18.4 ± 3.0	1.8 ± 0.3	16.3 ± 3.0	29.486 (<0.001) a > b > c
	moderate (23–41) ^b^		189(76.3)	23	41	30.9 ± 5.2	3.1 ± 0.5	12.7 ± 3.7	
	high (≥42) ^c^		14(5.6)	42	50	45.3 ± 2.6	4.5 ± 0.3	8.8 ± 3.3	
Secondary traumatic stress	total			10	41	22.8 ± 6.5	2.3 ± 0.7	-	-
	low (≤22)		136(54.8)	10	22	18.0 ± 3.1	1.8 ± 0.3	12.7 ± 4.0	−2.177 (0.030)
	moderate (23–41)		112(45.2)	23	41	28.7 ± 4.3	2.9 ± 0.4	13.8 ± 3.8	
burnout	total			10	45	28.3 ± 5.6	2.8 ± 0.6	-	-
	low (≤22) ^a^		33(13.3)	10	22	19.4 ± 2.5	1.9 ± 0.3	8.8 ± 2.9	31.888 (<0.001) a < b < c
	moderate (23–41) ^b^		211(85.1)	23	40	29.4 ± 4.2	2.9 ± 0.4	13.7 ± 3.6	
	high (≥42) ^c^		4(1.6)	42	45	43.8 ± 1.5	4.4 ± 0.2	18.5 ± 3.0	
**turnover intention**				4	20	13.2 ± 4.0	3.3 ± 1.0	-	-

*N* = 248, * GHQ coding 0011 cuf-off 10/11. ** GHQ coding 0123 cuf-off 23/24 or 27/28. *** effort–reward ratio = e/(r × c) c: defines a correction factor for different numbers of items in the nominator and denominator c = e/r = 6/11 = 0.545454.

**Table 3 healthcare-11-02695-t003:** Correlation between the research variables.

Variables	1	2	3	4	5	6	7	8
	r (*p*)	r (*p*)	r (*p*)	r (*p*)	r (*p*)	r (*p*)	r (*p*)	r (*p*)
1. psychological distress	1							
2. effort	0.447 ** (<0.001)	1						
3. reward	−0.376 ** (0.001)	−0.565 ** (<0.001)	1					
4. overcommitment	0.492 ** (<0.001)	0.557 ** (<0.001)	−0.381 ** (<0.001)	1				
5. compassion satisfaction	−0.323 ** (<0.001)	−0.194 ** (0.002)	0.253 * (0.024)	−0.160 ** (0.009)	1			
6. secondary traumatic stress	0.479 ** (<0.001)	0.368 ** (<0.001)	−0.325 ** (0.001)	0.535 ** (<0.001)	0.138 * (0.039)	1		
7. burnout	0.606 ** (<0.001)	0.515 ** (<0.001)	−0.534 ** (<0.001)	0.537 ** (<0.001)	−0.516 ** (<0.001)	0.558 ** (<0.001)	1	
8. turnover intention	0.345 ** (<0.001)	0.390 ** (<0.001)	−0.470 ** (<0.001)	0.354 ** (<0.001)	−0.530 ** (<0.001)	0.234 ** (<0.001)	0.578 ** (<0.001)	1

*N* = 248. * *p* < 0.05, ** *p* < 0.01.

**Table 4 healthcare-11-02695-t004:** Factors affecting participants’ turnover intention.

Variables	Model 1				Model 2					
	B	β	t	*p*	B	SE	β	t	*p*	VIF
(constant)	6.75		2.687	0.008	5.568	3.044		1.829	0.069	
gender	1.845	0.115	1.858	0.064	0.641	0.809	0.04	0.792	0.429	1.12
age of 20 group	0.738	0.093	0.621	0.535	0.553	0.96	0.07	0.576	0.565	6.53
age of 30 group	0.401	0.05	0.383	0.702	0.257	0.847	0.032	0.303	0.762	4.852
marital status *	−0.557	−0.069	−0.892	0.373	0.081	0.506	0.01	0.16	0.873	1.715
religion *	0.815	0.103	1.514	0.131	0.415	0.441	0.052	0.941	0.347	1.365
3-year college	0.762	0.083	0.724	0.47	0.457	0.868	0.05	0.527	0.599	3.919
4-year college	0.314	0.039	0.327	0.744	0.127	0.778	0.016	0.163	0.871	4.168
general ward	−2.212	−0.197	−2.893	0.004	−1.251	0.632	−0.112	−1.979	0.049	1.406
ICU	−0.282	−0.034	−0.446	0.656	−0.155	0.511	−0.019	−0.304	0.762	1.658
ER	−1.738	−0.188	−2.550	0.011	−0.364	0.562	−0.039	−0.648	0.518	1.623
staff nurse	1.63	0.193	1.301	0.195	0.85	1.012	0.1	0.84	0.402	6.307
assistant manager Nr	2.892	0.285	2.359	0.019	1.41	0.998	0.139	1.414	0.159	4.268
experience of wearing PPE during COVID-19	1.13	0.141	2.301	0.022	0.764	0.399	0.096	1.914	0.057	1.101
psychological distress **					0.973	0.46	0.123	2.115	0.036	1.492
effort-reward ratio					2.02	0.815	0.158	2.477	0.014	1.803
overcommitment					0.101	0.091	0.073	1.106	0.27	1.919
compassion satisfaction					−0.177	0.037	−0.349	−4.846	0	2.29
second traumatic stress					0.061	0.047	0.1	1.292	0.198	2.643
burnout					0.157	0.064	0.224	2.453	0.015	3.677
R^2^	0.179				0.484					
Adj. R^2^	0.133				0.441					
⊿Adj. R^2^	0.179				0.305					
F (*p*)	3.921(<0.001)				22.447 (<0.001)					

*N* = 248. Durbin–Watson = 2.049. B = unstandardized coefficients; β = standardized coefficients; SE = standard error; VIF = variance inflation factor; Adj. R^2^ = Adjusted R square; ICU = Intensive Care Unit; ER = Emergency Room. * dummy variable (0 = No, 1 = Yes). ** GHQ 23/24.

## Data Availability

Please contact the corresponding author for data availability.

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
