# Peer review of "An Exploratory Study of Psychological Distress, Professional Quality of Life, Effort-Reward Imbalance, and Turnover Intention of Hospital Nurses during the COVID-19 Pandemic"

_healthcare, 2023, doi:10.3390/healthcare11192695_

Round 1
Reviewer 1 Report
This is a well presented and useful article. I have only minor suggestions for revision.

My only issue is regarding the apparent interchange between the words 'empathy' and 'compassion'. See my notes in the text of the article about this.
Author Response
Thank you very much for taking the time to review this manuscript. Please find the detailed responses below and the corresponding revisions/corrections highlighted/in track changes in the re-submitted files

Reviewer 2 Report
In my opinion, the paper is sound and deserves to be published. However, some aspects should be improved:
page 2 (line 44): The meaning of the score 3.79 is not clear.
page 2 (line 60): When the authors refer to the ERI model, does this mean that this is the model you are using for the analysis in the paper?
The variables which are defined in the Introduction are not totally clear, in my opinion. For instance, is “empathy satisfaction” the same as “compassion satisfaction”?
Another example is “empathy fatigue”, which is said to be categorized as both burnout and secondary traumatic stress. Which categorization is used in this paper? Please, clarify this point.
Despite this vagueness in the Introduction, the method is clear and the results are well exposed in my opinion.
Author Response

(The authors gave the same response as above.)
